# Epithelial–Mesenchymal Transition Induced in Cancer Cells by Adhesion to Type I Collagen

**DOI:** 10.3390/ijms24010198

**Published:** 2022-12-22

**Authors:** Hitomi Fujisaki, Sugiko Futaki

**Affiliations:** 1Nippi Research Institute of Biomatrix, Ibaraki 302-0017, Japan; 2Anatomy and Cell Biology, Faculty of Medicine, Osaka Medical and Pharmaceutical University, Osaka 569-8686, Japan

**Keywords:** type I collagen, gel formation, epithelial–mesenchymal transition, tumor microenvironment, cancer cells

## Abstract

The epithelial–mesenchymal transition (EMT) is an important biological process that is physiologically observed during development, wound healing, and cancer invasion. During EMT induction, cancer cells lose their epithelial properties owing to various tumor microenvironmental factors and begin to exhibit mesenchymal properties, such as loss of apical–basal polarity, weakened intercellular adhesion, and promotion of single cell migration. Several factors, including growth factor stimulation and adhesion to type I collagen (Col-I), induce EMT in cancer cells. Cells adhere to Col-I via specific receptors and induce EMT by activating outside-in signals. In vivo, Col-I molecules often form fibrils, which then assemble into supramolecular structures (gel form). Col-I also self-assembles in vitro under physiological conditions. Notably, Col-I can be used as a culture substrate in both gel and non-gel forms, and the gel formation state of Col-I affects cell fate. Although EMT can be induced in both forms of Col-I, the effects of gel formation on EMT induction remain unclear and somewhat inconsistent. Therefore, this study reviews the relationship between Col-I gel-forming states and EMT induction in cancer cells.

## 1. Introduction

The mutation of genes in cancer cells often results in an abnormal growth program and excessive proliferation. Cancer cells influence and reorganize their surrounding microenvironments to form cancer tissue. In addition, proliferating cancer cells acquire migratory ability, leave the primary tumor, and begin invasion. Excessive proliferation and acquisition of migratory ability are hallmarks of cancer cells, and the surrounding tumor microenvironment (TME) significantly contributes to the acquisition and enhancement of these features. TME components including cancer cells interact with each other for cancer tissue maintenance, expansion, or cell dormancy [1,2,3]. Through pathological analysis, much has been learned in recent years about TME factors, such as growth factors, transcription factors, and extracellular matrix (ECM); however, a comprehensive understanding of TME remains unachieved owing to the diverse related factors and complex interactions between them [3]. Because cancer cells are thought to modify TME in a way that promotes their survival and proliferation, research on TME is also crucial for cancer treatment. TME comprises many components: multiple types of cells (cancer cells, cancer-associated fibroblasts (CAFs), immune cells, T cells, and macrophages), growth factors and cytokines [3,4], exosomes [5], microRNAs [6], and ECM including collagen [1,2,3]. Although correlations among these components have been reported, the roles of ECM remain unclear due to its diversity and variations of controlling factors. Type I collagen (Col-I) predominates among ECMs in TME and exerts complex effects on cells [7,8]. In this study, we focus on Col-I recognition and EMT induction in cancer cells.

Col-I is said to play a role in cancer malignancy for both progression and inhibition [9]. For the progression, excessive Col-I deposition, which is mainly produced by CAFs, on the tissue results in tissue hardening, making it a favorable scaffold for invasion [10]. Excessive ECM accumulation induces tissue fibrosis, creating a chronic inflammatory environment with activated CAFs. Fibrosis refers to tissue remodeling caused by excess ECM accumulation and cross-link formation in normal tissues. In addition to Col-I, fibrosis-induced tissues also contain collagens II, III, V, and IX [11,12], cross-linker glycoproteins (e.g., fibronectin (FN) and tenascins) [13,14], proteoglycans (heparan sulfate, CD44) [15,16], and others that interact with each other and are deposited. During the malignant transformation process in cancer, cell-produced growth factors such as transforming growth factor (TGF)-β, fibroblast growth factor (FGF), and hepatocyte growth factor stimulate increased production of ECM such as Col-I, which in turn promotes fibrosis. In addition, the excessive cross-linking of Col-I fibrils by various cross-linking agents, such as lysyl oxidase (LOX), increases tissue stiffness. ECM accumulation and cross-linking are thought to provide directionality to fiber bundles, the solid footing of cancer cells, and “migration highways” [17]. On the other hand, some researchers suppose that the aggregate structure of Col-I fibrils inhibits cell invasion by creating a barrier against (1) cancer cell migration into the stroma and (2) immune cell migration toward cancer tissue [9,18]. Based on this view, the activation of degradative enzymes such as matrix metalloproteinases (MMPs) produced by cancer cells during the subsequent malignant transformation process provides a pathway for cancer cell migration by degrading the barrier-functioning Col-I supramolecular aggregation. Furthermore, degraded Col-I fragments could act as chemotactic factors [3]. 

The roles of Col-I in cancer progression are not only acting as physical structures as scaffolds or barriers to migration. In the past, ECM such as Col-I was thought to function only as an inert scaffold, but recent studies have shown that cells attach to scaffolds via cell surface receptors and that receptor-mediated signaling occurs within cells, thereby influencing cell behaviors [19]. As we will discuss in this paper, when epithelial cancer cells adhere to the stromal ECM, Col-I, EMT may occur, in which the cells acquire mesenchymal cell characteristics. 

Understanding complex biological phenomena requires the use of simplified in vitro experiments. In culture experiments, gel-like scaffolds, which are significantly softer than “hard” scaffolds made of glass or plastic, are thought to more closely resemble the physiological environment [20]. The experimental studies on induction of EMT began in the 1980s in embryology with the use of Col-I gel culture systems [21], and since then, various methods, such as growth factor stimuli and modulation of culture substrate stiffness, have been investigated. Cell biology, cancer biology, pharmacology, and developmental biology have utilized Col-I gel culture systems, and the development of Col-I-based culture systems continues to this day [22,23]. Researchers can alter the thickness, stiffness, and length of fibrils in Col-I gels by regulating collagen solution concentration, incubation temperature during gel preparation, and pH [22]. As a result, the state of gel formation may affect cells adhering to Col-I gels. Additionally, various modified Col-I gel culture systems changing gel stiffness and surface morphology can be utilized [23].

Some reports on EMT induction by adhesion to Col-I include insufficient information on the substrate preparation despite the critical role of the gel formation state of Col-I in culture substrate. The lack of a precise description of the Col-I gel formation states may have rendered the correlation between EMT induction and Col-I effect unclear. Therefore, we believe that sorting out the findings regarding EMT induction and adhesion to Col-I gels would contribute to future cancer research, and we have summarized previous studies. As an example, we also have discussed the findings on the lung cancer cell line A549. Figure 1 depicts a schematic diagram of the Col-I roles dependent on their forms in TME and the induced cell behavior. 

## 2. Col-I Fibril Formation and Cell Recognition

### 2.1. Collagen Overview

Collagens are a protein superfamily characterized by the helical association of three collagen α polypeptide chains to form a trimer. Currently, 28 collagen types have been reported and classified according to their modes of supramolecular structure formation, such as fibrillar, network-forming, and FACIT (fibril-associated collagens with interrupted triple helices) types. Each type of collagen molecule is regularly arranged to form supramolecular aggregates of a unique shape. Col-I, which is the main focus of this study, is a fibrillar-type collagen. In the case of fibrillar-type collagen, collagen molecules form fibrils, and further fibrils aggregate into bundles (fibers). Through collagen–collagen interaction, cross-link formation, and interaction with other ECMs, these supramolecular structures contribute to the physical structure properties that regulate the tissue’s mechanical strength [24,25]. 

### 2.2. Col-I

Col-I, a fibrillar collagen, is most abundant and ubiquitous in the interstitium. Most Col-I molecules form fibrils and fibril-associated structures in vivo. The method for purifying Col-I has long been established [26], and many experiments and applications have been reported. In vitro, Col-I molecules reassemble regularly to form fibrils. Col-I fibrils further assemble to form supramolecular, gel-like three-dimensional structures (gels) [24,25,27,28,29]. Cells utilize distinct adhesion receptors depending on whether or not Col-I gels are formed [25,29]. The Col-I molecule has multiple cell recognition sites for integrins, which comprise particular sequences of amino acids, such as glycine-phenylalanine (leucine, methionine)-hydroxyproline-glycine-glutamic acid-arginine (GF(L, M)OGER) and similar sequences. When Col-I molecules form triple helical structures, these sequences serve as recognition sites [30]. 

### 2.3. Col-I and Gelatin

Gelatin refers to collagen molecules whose triple helical structure has been destroyed by thermal denaturation [31]. Gelatin also has cell adhesion activity. Its activity is independent of the triple helical structure of Col-I but depends on the amino acid sequence of the collagen peptide chain, arginine-glycine-aspartic acid (RGD), and adhesion is mediated by RGD-type receptors, such as integrins α5β1 and αvβ3. The RGD sequence is absent on the molecule surface when Col-I molecules form a triple helix. Therefore, when gelatin is used as a substrate in a culture system, cell adhesion differs from that of Col-I. Some cells, such as HT1080, have higher adhesion activity to Col-I than to gelatin [32], but in some other cases, vice versa. For example, U937 cells induced to differentiate into macrophage-like cells have a low affinity for intact Col-I and a high affinity for heat-denatured Col-I (i.e., gelatin) [33]. Macrophage-like U937 cells on gelatin form large cell colonies [34], increase ROS levels, enhance autophagy, and promote inflammatory mediator release [35]. These experiments suggest that immune cells, other epithelial cells, and fibroblasts have different Col-I form-dependent adhesion, i.e., that each cell has a different role in the organism depending on the form of Col-I.

Various proteases degrade collagen polypeptide chains with disrupted triple helical structures into smaller peptides with relative ease. Collagen peptides, once degraded to the size of a few amino acids, serve more as soluble factors than as scaffolds for cell adhesion. Specific amino acid sequences of Col-I degradation products, such as proline-hydroxyproline (PO) and glycine-proline-hydroxyproline (GPO), can regulate physiological activities such as cell proliferation and chemotaxis [36,37]. Recently, the mechanism of integrin β1-mediated intake of collagen peptide, PO, in tenocytes and its effect on cell behavior has been reported [38].

### 2.4. Col-I Gel Formation States and Cell Recognition via Collagen Receptors

The major collagen receptors expressed in various cells that directly recognize Col-I are integrin α1β1, α2β1, α10β1, α11β1, and the discoidin domain receptor (DDR) family, DDR1 and DDR2. Other receptors, including GPVI, OSCAR, LAIR-1, and GPR56, are cell specifically expressed [39]. Integrins are transmembrane receptors that function in heterodimers comprising a specific combination of 18 different α subunits and 8 different β subunits. The heterodimeric combination of integrin α and β subunits is fixed, and ECM recognized by each integrin heterodimer is specific [40]. In general, cells express multiple types of integrins and use them differently, depending on the substrate state. Integrin α1β1, α2β1, α10β1, and α11β1 that recognize Col-I have ligand specificity depending on the status of collagen fibril supramolecular structure. Col-I fibrils are recognized by integrins α2β1 and α11β1 but not by integrins α1β1 and α10β1. This is because, when fibrils are formed, some of the recognition sites on the surface of Col-I molecules are incorporated into the fibril and no longer exist on the surface [29].

By forming fibrils, Col-I may influence cell signaling via integrins in at least three points. The first point is the integrin recognition site location on the Col-I supramolecular structure surface [29], and the second is the cell recognition of Col-I surface topography [41]. The third point is the sensing stiffness (or elastic modulus) ability of integrins. As integrins are mechanical sensors that sense the adhered substrate stiffness, adherent cells are affected by the scaffold stiffness [39,41]. These factors of Col-I (presence of the recognition site, surface topography, and stiffness) are all inseparable and simultaneously changed according to the gel-forming state of Col-I. For these reasons, experimental results should be discussed with caution when Col-I is used as a culture substrate.

The supramolecular aggregate structure of Col-I gels in a culture system can be confirmed by observation using a scanning electron microscope [42]. However, verifying the state of fibril formation on the dish surface after applying a low concentration of Col-I can be difficult. Col-I molecules at low concentrations on the culture dish surface can form fibrils under physiological conditions and may form thin fibers, but they do not form gels during cell culture. In other words, when cells are cultured on the non-gel form of Col-I, integrins may sense both the stiffness of the culture dish and Col-I with an indeterminate fibril formation state. Complexity in the formation state of Col-I fibrils may contribute to the difficulty in assessing its impact on cellular behavior. Figure 2 shows a schematic diagram of the factors related to the Col-I gel formation state that influence signal transduction via integrins. Table 1 lists the characteristics of the Col-I gel and non-gel forms. Stiffness is indicated by elasticity.

DDR is a receptor tyrosine kinase with a homology domain to the discoidin domain and is classified into two categories, DDR1 and DDR2 [43]. DDR is involved in cell proliferation, tissue fibrosis, and cancer invasion via the remodeling of ECM. Both DDR1 and DDR2 bind to collagen fibrils as ligands, but differences exist in the specificity of collagen type recognition. DDR1 is expressed mainly in epithelial cells and leukocytes, whereas DDR2 is expressed in fibroblasts and smooth muscle cells [44]. Although the signals activated via DDR are independent of integrins [45], they may act in concert [46]. 

**Table 1 ijms-24-00198-t001:** Characteristics of the gel-forming state of Col-I.

Col-I Gel Formation States	Elasticity (Pa) [47]	Topography [41]	Integrins for Recognition [29]
Col-I gel form (fibrils)	10^2^~10^3^	fibrous	Integrin α2β1, α11β1
Col-I non-gel form (on polystyrene)	>10^9^	flat	Integrin α1β1, α2β1, α10β1, α11β1

## 3. Diverse EMT-Inducing Factors

### 3.1. EMT Overview

EMT is a concept that originated in embryology [21]. During development, cells migrate and rearrange themselves in tissues due to various factors and reconstitute new tissues. A high migratory capacity is an important EMT characteristic. The importance of EMT and its reverse process, mesenchymal–epithelial transition (MET), has been demonstrated in developmental processes, such as gastrulation and neural crest cell migration [48]. Subsequently, it has been found to be a widespread phenomenon in events other than developmental stages, such as wound healing [49], cancer invasion, and metastasis [50], where cells migrate from their original location in living organisms. During wound healing in the skin, inflammation and injury cause EMT in keratinocytes, which initiates tissue repair. A duration that EMT continues in this process divide wound healing from fibrosis. If EMT occurs in a timely manner and at an appropriate length, the wound will heal and return to healthy tissue, but if EMT persists longer, the cells will continue to produce ECM and the tissue will undergo fibrosis [49]. In the case of cancer invasion, EMT is considered an important process that operates when epithelial cancers metastasize to the interstitium. Dynamic cellular changes are a fascinating topic for researchers, and the molecules and mechanisms involved have been vigorously elucidated. There are still numerous reports and excellent reviews covering a wide range of EMT topics from a variety of perspectives. 

### 3.2. Various Factors for EMT Induction

EMT-inducing factors are diverse [51]. One of these is growth factors such as TGF-β. In addition, each normal tissue has an appropriate “stiffness”, and pathological changes can occur when a tissue’s normal inherent stiffness is altered. When fibrotic tissue becomes stiffer than normal tissue [52], the increased stiffness of the cell adhesion scaffold alone induces EMT [53]. Furthermore, as discussed below, EMT is induced by adhesion to specific ECMs, such as Col-I, with signaling activated by the receptor.

Among growth factors, the well-studied TGF-β family is a potent EMT inducer. In addition to causing EMT induction in many cancer cells, its action also activates multiple signaling pathways associated with cancer malignancy, including fibrosis, immunosuppression, and angiogenesis enhancement. Therefore, TGF-β could promote the malignant transformation of cancer. Simultaneously, TGF-β has been reported to inhibit the proliferation of some types of cancer cells and induce apoptosis of early cancers, and from this perspective, it is said to have an inhibitory effect on cancer malignant transformation [54]. Thus, although the effect of TGF-β on malignant transformation is two-sided and fraught with paradoxes [55], genetic mutations in the TGF-β signaling system have been reported in many cancers [56,57] and undoubtedly play an important role in determining cancer cell behavior. Stimulation of the TGF-β family activates two TGF-β receptors on the cell surface (TGF-β receptor I and II), which in turn activate two largely independent pathways, the Smad and non-Smad pathways [58]. In the Smad pathway, signals are transmitted through multiple Smads. In the cell, various Smads are sequentially phosphorylated, and the phosphorylated Smad complexes are finally transferred into the nucleus, where they form complexes with various transcription factors and transcription coactivators to regulate the transcription of the target genes. More than 100 Smad-binding factors have been reported, and their regulation is complicated. Besides, the non-Smad pathway activates diverse signaling systems, including p38 mitogen-activated protein kinase (MAPK), p42/p44 MAPK, c-Src, mammalian target of rapamycin, Rho A, RAS, phosphoinositide 3-kinase (PI3K)/Akt, protein phosphatase 2A/p70S6K, and JNK MAPK, suggesting cross-talk of these pathways [59,60,61,62]. As a result, actin stress fiber remodeling by Rho A activation, suppression of epithelial marker expression by regulating transcription factor expression, and increased expression of mesenchymal markers are induced, leading to ECM remodeling, weakening of intercellular adhesion, and changes in cell morphology.

### 3.3. EMT Markers

Epithelial cells express intercellular adhesion molecules, such as E-cadherin, claudin, and occludin, whereas mesenchymal cells express mesenchymal markers, such as vimentin, FN, and N-cadherin, and have weak intercellular adhesion [63]. EMT markers include many factors, such as specific transcription factors, ECM production, cell migration activity, cell morphology, and other cell behaviors. Therefore, it is difficult to examine and report all of them in a single paper. EMT markers are not necessarily expressed in correlation with each other.

Cadherins, for example, are intercellular adhesion molecules that exist in cell membranes. They form a superfamily with subtypes such as epithelial (E-), neural (N-), and retinal (R-) types of cadherin, which bind homophilically to form intercellular adhesions, respectively [64,65]. In general, epithelial cells express more E-cadherin, and mesenchymal cells express more N- and R-cadherins. In epithelial cells, E-cadherin forms adherence junctions and maintains apical–basal polarity and epithelial integrity. During organogenesis and cancer invasion, the emergence of N-cadherin-positive cells that migrate from the epithelial tissue in which epithelial cells adhere to each other via E-cadherin may be observed, and this phenomenon is considered one of the major landmarks of EMT. The decrease in E-cadherin expression is paralleled by an increase in N-cadherin expression, which is referred to as the cadherin switch. Decreased expression of E-cadherin [66] and cadherin switch [67] are considered key indicators of EMT. E-cadherin decreasing and N-cadherin increasing do not necessarily occur simultaneously, as the regulation of the decreased expression of E-cadherin and the increased expression of N-cadherin are mediated through independent signaling systems. An example is the transcriptional regulation of EMT markers by the transcription factors, δEF (ZEB1) and Smad-interacting protein 1 (SIP1) [68]. Double knockdown of δEF1 and SIP1 by siRNA in a mammary epithelial cell line almost completely suppressed TGF-β-stimulated suppression of E-cadherin expression. However, knockdown of SIP1 and δEF1 has no effect on N-cadherin expression. Other E-cadherin transcription regulators include Zn finger-type transcription factors such as Snail and Slug, and many basic helix-loop-helix (bHLH)-type transcription factors such as E2A and Twist [69].

### 3.4. Various EMT-Related Transition

Over the past two decades, EMT research has flourished, and a variety of cell behaviors have been reported. Accumulating evidence has clarified that two or more phenomena associated with EMT and captured as equally induced properties may not always occur simultaneously. For example, in 2015, two groups each established model systems to observe the course of EMT and cell invasion over time in mice [70,71]. Their results clearly showed that the conventional idea that EMT induction weakens intercellular adhesion and promotes the migration and invasion of single cells may not be simultaneously valid. The reports of these two groups suggest that EMT induction is not necessary for cancer metastasis but is effective for drug resistance. In addition, reports of EMT-related conversions, partial EMT, which exhibits mixed and intermediate epithelial and mesenchymal properties [72,73,74], and collective migration are increasing [74]. The EMT International Association has recently published a consensus statement to provide researchers with guidelines in light of diverse reported phenotypes and numerous EMT-related factors [75].

## 4. EMT Induction by Adhesion to ECM 

### 4.1. EMT Induction Regarding Col-I

Signal activation through collagen adhesion influences EMT induction processes such as cadherin switch and cell morphology change [46,64,76,77]. Epithelial cells typically maintain polarity by adhering to laminin (LM), a component of the basement membrane, via LM receptors, integrins α3β1, α6β1, and α6β4 [78]. However, Col-I-contacting cells adhere via collagen receptors [39]. LM receptor α3β1 and collagen receptor α2β1 activate distinct signaling pathways. For example, in human foreskin keratinocytes, integrin α3β1, which recognizes LM 332, activates Rho-GTPase, and integrin α2β1, which recognizes Col-I, activates a Rho-dependent signal that causes actin cytoskeletal reorganization [79]. Notably, the signal activated by unusual adhesion to the stromal ECM may cause the epithelial cells to lose their epithelial nature and acquire stromal cell-like properties.

To identify the signaling pathways involved in EMT induction regarding Col-I, Shintani and colleagues used a short hairpin RNA technique to knock down integrin β1 and DDR1 expression in pancreas cancer line BxPC3 cells and successfully suppressed the induction of the cadherin switch through adhesion to Col-I [46]. Knocking down integrin β1 alone did not suppress the EMT. The involvement of focal adhesion kinase (FAK) downstream of activated integrin and protein tyrosine kinase 2 (pyk2) downstream of DDR1 in activation was demonstrated, and inhibitor screening was conducted to investigate further downstream signaling. The results suggest that JNK1 and its downstream c-jun activation are involved in increased N-cadherin expression. It is a Col-I-induced EMT, but the detailed Col-I treatment conditions of the adhesion scaffolds were not described in this report, so how Col-I is recognized by the receptors remains unclear. Koenig et al. reported that integrin β1 and FAK-mediated regulation reduces E-cadherin expression in pancreatic cancer-strain cells cultured on molecular Col-I. The Col-I treatment concentration was 10 μg/mL or 5 μg/cm^2^ [76].

### 4.2. EMT Induction Regarding FN 

Adhesion to ECMs other than Col-I can result in an EMT-like phenomenon. FN is a 440 kDa dimeric glycoprotein that enhances the migratory ability of cells [80]. FN molecules are multidomain proteins that bind to many molecules, including Col-I, gelatin, and growth factors, and form aggregates with other FNs to serve as scaffolds for cell adhesion. The state of FN supramolecular aggregate formation affects cell behavior [81]. FN is also found in large amounts as a soluble form in plasma, where it functions at the site of tissue injury. During wound healing, for example, FN promotes the conversion of fibroblasts to myofibroblasts [82]. Myofibroblasts are similar to certain CAF subtypes. FN expression has been reported to inversely correlate with E-cadherin expression during the malignant transformation of cancer [83]. However, in the experimental systems of Shintani et al. [46] and Koenig et al. [76], adhesion to FN-coated culture dishes had no effect on EMT induction. The specific receptor that binds to FN is integrin α5β1 [40]. FN is a mesenchymal marker in EMT induction experiments [48,68].

### 4.3. EMT Induction Regarding LM

LM is a basement membrane protein that forms a heterotrimer molecule composed of α, β, and γ chains, forming a superfamily of more than 12 types. LM also has different bioactivities in molecules, supramolecular aggregates, and fragmented peptides. In epithelial cells, adhesion to LM promotes cell migration and maintains epithelial properties, such as apical–basal polarity and intercellular adhesion structures [84]. Notably, increase of LM can serve as a malignant cancer marker; the LM332 expression level can rise when pancreatic ductal adenocarcinoma (PDAC) becomes malignant [85]. In this report, knockdown of the LM β3 chain enhances epithelial properties, as indicated by an increase in E-cadherin levels and a decrease in vimentin levels. Based on this decreased expression of vimentin, the authors believe that the LM β3 chain is an EMT inducer. LM γ2 chain also affects cell proliferation, apoptosis, invasion, and migration; inhibition of LM γ2 chain expression in pancreatic cancer cell lines can increase cell sensitivity to drugs and induce apoptosis while inhibiting colony-forming capacity, migration, and invasion ability [86]. On the other hand, as described above, there is a report that epithelial cells maintain polarity by adhering to LM [78]. Further detailed studies are needed.

## 5. EMT Induction in A549 Cells on Col-I 

We have shown in Section 3 that EMT in cancer cells can be induced by diverse factors, including growth factor stimulation and adhesion to the ECM, such as Col-I, and that the cellular responses may deviate from the classical definition of EMT. The lung carcinoma cell line A549 cells has been reported in numerous EMT induction experiments. EMT induction in A549 cells often employ growth factor stimulation, especially TGF-β treatment [87,88,89]. In addition, although FGF2 treatment alone does not cause significant EMT-related changes, synergistic effects have been reported in combination with TGF-β1 for the promotion of EMT induction [90]. The adhesion of A549 cells to Col-I also elicits EMT induction [91,92]. In these reports, Col-I treatment conditions vary, as do the cellular responses and the proposed mechanisms of EMT induction. In this section, we summarize the studies of EMT induction in A549 cells to exemplify the diversity involved in EMT induction by recognizing Col-I.

### 5.1. EMT Induction by TGF-β Autocrine

Shintani et al. showed that adhesion to Col-I induced a cadherin switch and promoted single cell scattering in A549 cells [91]. They report the different EMT-inducing mechanisms in A549 and pancreatic cancer cells. In the case of pancreatic cancer cells, cooperative activation of FAK and pyk2 via integrin β1 and DDR1, respectively is an important EMT-inducing pathway [46]. PI3K and extracellular signal-regulated kinase pathways are activated to produce TGF-β3 by adhesion to Col-I, and EMT is induced by TGF-β autocrine in A549 cells [91]. Col-I treatment of adhesive scaffolds is not described in detail in the report.

### 5.2. EMT Induction on Col-I-Coated Polymer Gels 

Shukla et al. designed an experimental system comprising Col-I-coated polydimethylsiloxane (PDMS) substrates with varying stiffnesses to investigate the correlation between the stiffness of gels containing Col-I and EMT induction in A549 cells [92]. The gel surface of PDMS is flat and smooth. They examined cell migration parameters and cadherin switching on PDMS gels coated with Col-I with different stiffnesses. In their experiments, migration parameters are substrate-stiffness dependent, but substrate stiffness do not affect the cadherin switch. Interestingly, TGF-β treatment cause a similar degree of EMT induction, regardless of substrate stiffness.

### 5.3. EMT Induction in Col-I Gel Sandwich Culture with TGF-β Treatment 

The behavior of A549 cells treated with TGF-β1 and grown in sandwich culture on Col-I gel has been described [93]. Since there have been reports of crosstalk between signals activated by TGF-β receptors and integrins [94], it is possible that these two signaling systems are also interacting in the Col-I experimental system. Although many conditions differ and cannot be easily compared to other experiments, the findings of this report [93] on the combined scaffold and soluble factor experiments are important. For EMT induction, the authors compare the cellular response to 2 different Col-I substrates, on Col-I molecule and in sandwich cultures with Col-I gels. Col-I sandwich gel cultures have more physiological conditions than on Col-I molecule. In both culture methods, A549 cells showed increased expression of invasion markers, LM γ2 chain, and MT1-MMP, in addition to various EMT markers. However, differences in migration and cell proliferation occurred between these two culture methods, with the sandwich method suppressing proliferation and migration. For the sandwich culture, the concentration of Col-I gel was 2.1 mg/mL, and the cell suspension containing Col-I was overlaid on top of the pre-formed Col-I gel.

### 5.4. Comparison of A549 Cell Behaviors between Col-I on Gel Culture and TGF-β1 Treatment 

We are particularly interested in the effect of the presence or absence of the Col-I fibril supramolecular structure on EMT induction. We cultured A549 cells on a non-gel form of Col-I (10 μg/mL), on Col-I gels (1 mg/mL), without TGF-β1, and compared with the same A549 cells cultured on an untreated culture dish surface without TGF-β1 (as an EMT-induced negative control) and with TGF-β1 (as a positive control) [95]. A549 cells on an untreated culture dish surface form cell clusters via E-cadherin, whereas the cells on the non-gel form of Col-I show weaker intercellular adhesion and higher single-cell migratory capacity than those on an untreated culture dish. Cadherin switch is induced in cells on the non-gel form of Col-I. It is also induced on Col-I gels, but unexpectedly, the cells on the gels form cell clusters via N-cadherin. TGF-β1-treated cells develop fibroblastic morphology and enhance migratory ability. TGF-β1-treated cells express high levels of EMT marker, vimentin, and increased expression of integrin α2 and β1, whereas cells cultured on Col-I either on non-gel or on gels do not express vimentin, and integrin α2 and β1 expression levels are not enhanced. This indicates that the EMT-like changes induced by TGF-β1 treatment differ from those induced by adhesion to Col-I. Adhesion to Col-I gels causes cadherin switch and enhances cell migration, but N-cadherin-mediated cell cluster formation deviates from the definition of EMT. We believe that many facts regarding EMT-related alterations in cell behavior, such as signal transduction pathways on Col-I, remain to be discovered. 

## 6. Conclusions

As mentioned above, it is becoming increasingly recognized that adhesion to Col-I has a significant impact on the cancer cell behavior, and many variations in EMT-related cell behavior due to Col-I recognition have been reported. For example, in pancreatic and lung tumor tissues, where Col-I tends to accumulate in the surrounding tumor area, fibrosis is a major contributor to malignant transformation and poor prognosis [49,52]. However, very recently, Nature published an article in which fibrosis inhibited malignant transformation, and activation of DDR1 signaling by collagen degradation products correlated with poor prognosis in pancreatic cancer [96]. The authors focused on the phenomenon in which MMPs expressed by pancreatic cancer determine the amount of collagen degradation products and correlate with poor prognosis, and they created mice expressing collagen that are resistant to MMP degradation. When cancer cells were transplanted into these mice, cancer cell growth was significantly inhibited. In addition, they demonstrated that DDR1 activation triggered inflammatory signaling and that inhibiting this signal transduction suppressed cancer cell growth. Interestingly, undegraded Col-I also bound to DDR1, but DDR1 was degraded by ubiquitination, thereby preventing inflammatory signal activation. In other words, undegraded Col-I inhibits the malignant transformation of cancer cells. In this experimental system, degraded or non-degraded Col-I has opposite effects on malignant cancer transformation. These results are quite interesting when considering the role of Col-I in TME. 

As described in this report, while significant progress has been made in recent years to elucidate the role of Col-I in TME, the contribution of Col-I to cancer malignancy is proving to be more complex, indicating that additional research is required to clarify the mechanism by which adhesion to Col-I induces EMT. Further research will likely yield new therapeutically applicable findings. In this report, brief descriptions of each issue are provided. For more insight, please consult the individual review articles.

## Figures and Tables

**Figure 1 ijms-24-00198-f001:**
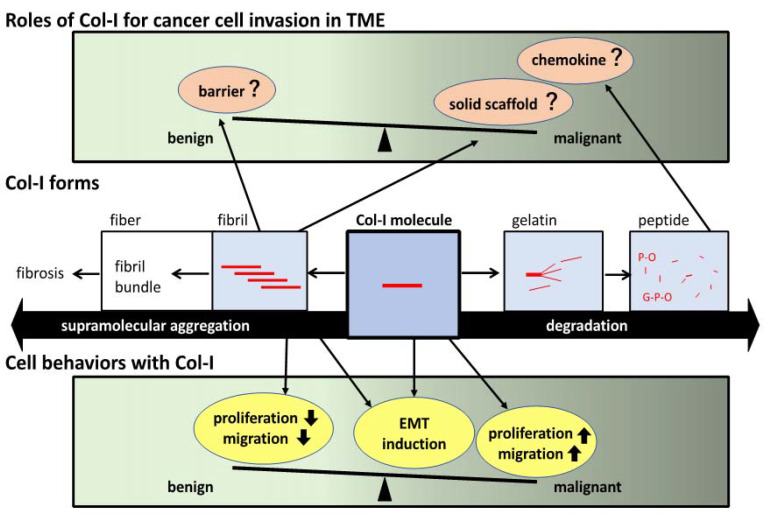
Schematic view of the roles of Col-I in cell invasion in TME. The middle row describes the type I collagen (Col-I) multiple forms. The (**top row**,**bottom row**) indicate the roles of Col-I in cancer cell invasion in tumor microenvironment (TME) and Col-I-influenced cell behaviors, respectively. Col-I molecules form supramolecular aggregates such as fibrils and fibers, or conversely, it forms gelatin upon denaturation. Gelatin is further degraded to peptides, which act as soluble factors (**middle row**). These variations in the Col-I form have distinct effects on cell behavior. In general, Col-I in the supramolecular aggregation state suppresses cell proliferation and migration, thereby inhibiting the malignant transformation of cancer, whereas Col-I molecules promote proliferation and migration, thereby promoting malignant transformation. In both conditions, EMT is induced (**bottom row**). It is said that Col-I, in its supramolecular aggregation state, exhibits contradictory functions: one is to act as a physical barrier against cancer and inflammatory cells, whereas the other is to provide a solid scaffold that promotes cancer cell invasion. Col-I peptides can act as chemotactic agents and may promote malignant transformation (**top row**).

**Figure 2 ijms-24-00198-f002:**
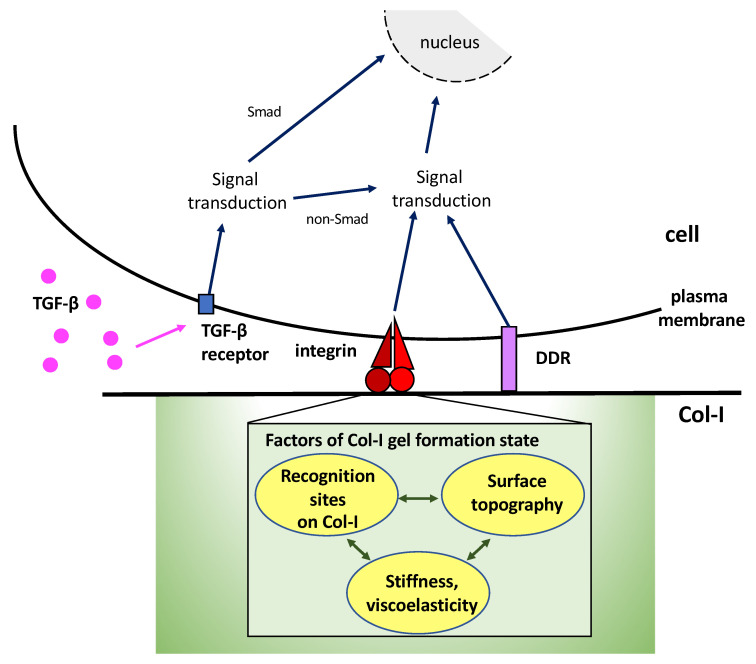
Effectors of integrin-mediated recognition on Col-I. Signaling activated by the recognition of Col-I substrates by integrins and DDR, as well as stimulation of soluble factors such as TGF-β, regulate cell behavior in concert. In addition, integrin-mediated signals are influenced by the factors on the gel-forming state of Col-I, such as the arrangement of cell recognition sites, surface topography, and stiffness.

## Data Availability

Not applicable.

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
