# Peer review of "Epithelial–Mesenchymal Transition Induced in Cancer Cells by Adhesion to Type I Collagen"

_ijms, 2022, doi:10.3390/ijms24010198_

Round 1
Reviewer 1 Report
The manuscript entitled “Description of Epithelial to Mesenchymal Transition Which Induced in Cancer Cells by Adhesion to Type I Collagen” by Hitomi Fujisaki et al. carefully reviewed the Epithelial to Mesenchymal Transition Which Induced in Cancer Cells by Adhesion to Type I Collagen. The premise of the work is very interesting, however in its present version, the manuscript requires several significant areas of improvement before consideration for publication.
In my opinion major improvements are required
1) Figure legends should be self-explanatory and properly abbreviated. More explanation is necessary.
2) Title: Description of Epithelial to Mesenchymal Transition Which Induced in Cancer Cells by Adhesion to Type I Collagen, I think a more appropriate title should be found.
3) I wonder if authors would provide a box (containing some bullet points) addressing some major points/ mechanisms/ challenges and/ or answers of some demanding questions of the discussed area.
4) There are some points either discussed haphazardly or overlooked, need to be discussed properly.
5) The quality of the figures is not good. It must be improved.
6) The text needs careful proof reading. There are many grammar and spelling mistakes.
7. Line117. the dot after ECMs is unnecessary. `formation, and interaction with other ECMs. [24,25].`
8. Line 124, there is a mistake in the line.
Author Response
Response to Reviewer 1 Comments
The manuscript entitled “Description of Epithelial to Mesenchymal Transition Which Induced in Cancer Cells by Adhesion to Type I Collagen” by Hitomi Fujisaki et al. carefully reviewed the Epithelial to Mesenchymal Transition Which Induced in Cancer Cells by Adhesion to Type I Collagen. The premise of the work is very interesting, however in its present version, the manuscript requires several significant areas of improvement before consideration for publication.
Thank you for your criticism. We have carefully read your comments and rewrote the manyuscript. Your points have helped us to improve the manuscript.
Point 1: Figure legends should be self-explanatory and properly abbreviated. More explanation is necessary.
Response 1: Thank you very much for your kind suggestions. We have added more explanations to figure legends.
Point 2: Title: Description of Epithelial to Mesenchymal Transition Which Induced in Cancer Cells by Adhesion to Type I Collagen, I think a more appropriate title should be found.
Response 2: Thank you very much for your professional advices. We have simplified the title.
Point 3: I wonder if authors would provide a box (containing some bullet points) addressing some major points/ mechanisms/ challenges and/ or answers of some demanding questions of the discussed area.
Response 3: Thank you very much for your helpful suggestion. We have added a few summary sentences after each section. We hope this change makes it easier to understand the main points.
Point 4: There are some points either discussed haphazardly or overlooked, need to be discussed properly.
Response 4: The role of type I collagen in tumor microenvironment is diverse, so, in this manuscript, we have focused on EMT induction. With your advice, we have refined the manuscript and inserted some texts to explain the relevance where the relationship was unclear.
Point 5: The quality of the figures is not good. It must be improved.
Response 5: Thank you very much. We rewrote figures.
Point 6: The text needs careful proof reading. There are many grammar and spelling mistakes.
Response 6: The manuscript has improved by a professional English proof reading service.
Point 7: Line117. the dot after ECMs is unnecessary. `formation, and interaction with other ECMs. [24,25].`
Response 7: Thank you for pointing this out. We have removed the period.
Point 8: Line 124, there is a mistake in the line.
Response 8: Thank you for pointing this out. We have corrected our mistake according to your helpful suggestions.

Reviewer 2 Report
The authors summarized current understandings of the relationship between EMT induction and adhesion to Col-I gels, which filled in the gap in the field. In general, the manuscript is well-written with sufficient background knowledge introduction in each section and figures are accurate and interpretable. The manuscript might benefit from adding a few summary sentences after each section, especially for section 4.1, 4.2, 4.3 which are the key parts of the review to better emphasize the points of view.
Author Response
Point: The authors summarized current understandings of the relationship between EMT induction and adhesion to Col-I gels, which filled in the gap in the field. In general, the manuscript is well-written with sufficient background knowledge introduction in each section and figures are accurate and interpretable. The manuscript might benefit from adding a few summary sentences after each section, especially for section 4.1, 4.2, 4.3 which are the key parts of the review to better emphasize the points of view.
Response: Thank you very much for your helpful advices. We have added a few summary sentences after section 2-5. We hope this change makes it easier to understand the main points.